# Prevalence and Transmission of Extended-Spectrum Cephalosporin (ESC) Resistance Genes in *Escherichia coli* Isolated from Poultry Production Systems and Slaughterhouses in Denmark

**DOI:** 10.3390/antibiotics12111602

**Published:** 2023-11-08

**Authors:** Meiyao Che, Tina Birk, Lisbeth Truelstrup Hansen

**Affiliations:** 1National Food Institute, Technical University of Denmark, 2800 Kongens Lyngby, Denmark; meic@food.dtu.dk; 2Department of Technology, University College Copenhagen, 2300 Copenhagen, Denmark; tibi@kp.dk

**Keywords:** antibiotic resistance, poultry production systems, slaughterhouses, ESC-resistant *E. coli*

## Abstract

The emergence of extended-spectrum cephalosporin (ESC)-resistant *Escherichia coli* is a global concern. This study aimed to assess the prevalence and transmission of ESC-resistant *E. coli* in the Danish broiler production system. Samples from two vertically integrated Production Systems (1 and 2) and two slaughterhouses (A and B) were analyzed (n = 943) for the occurrence of ESC-resistant *E. coli* from 2015 to 2018. ESC-resistant *E. coli* isolates were whole-genome sequenced (WGS) for characterization of the multi-locus sequence type (MLST), antibiotic resistance genes, virulence genes, and plasmid replicon types. An ad hoc core genome (cg) MLST based on 2513 alleles was used to examine the genetic relatedness among isolates. The prevalence of ESC-resistant *E. coli* in the conventional Production System 1 was 2.7%, while in Production System 2 the prevalence was 26.7% and 56.5% for samples from the conventional and organic production, respectively. The overall prevalence of ESC-resistant *E. coli* in broiler thigh and fecal samples ranged from 19.3% in Slaughterhouse A to 22.4% in Slaughterhouse B. In total, 162 ESC-resistant *E. coli* were isolated and shown to belong to 16 different sequence types (STs). The most prevalent STs were ST2040 (n = 85) and ST429 (n = 22). Seven ESC resistance genes were detected: *bla*_CMY-2_ (n = 119), *bla*_TEM-52B_ (n = 16), *bla*_CTX-M-1_ (n = 5), *bla*_TEM-52C_ (n = 3), *bla*_CTX-M-14_ (n = 1), *bla*_SHV-12_ (n = 1), and up-regulation of ampC (n = 16), with an unknown resistance gene in one isolate (n = 1). The carriage of *bla*_CMY-2_ in 119 isolates was primarily associated with IncI1 (n = 87), and IncK plasmids (n = 31). Highly similar *bla*_CMY-2_ carrying *E. coli* isolates from ST429 were found in production systems as well as in slaughterhouses. In conclusion, findings from this study indicate that ESC-resistant *E. coli* are transferred vertically from farms in the production systems to slaughterhouses with the potential to enter the food supply.

## 1. Introduction

Extended spectrum cephalosporins (ESC) resistance has been detected worldwide in bacteria associated with food production animals, an observation which may be related to veterinary use of cephalosporins [1]. The potential dissemination of ESC-resistant bacteria between the microbial reservoirs of food production animals and humans has been identified as a problem, especially as it may interfere with clinical options for treating *Enterobacteriaceae* infections in humans [2]. In Denmark, high levels of ESC-resistance were identified in poultry, leading to the implementation of a voluntary ban on the usage of cephalosporin in poultry production in 2010 [3]. Despite the ban on use of cephalosporins in the European Union, persisting ESC-resistant bacteria in the poultry industry have remained an issue in Denmark and other European countries [4]. In France, Casella and colleagues found that the prevalence of ESC resistance among *E. coli* in retail chicken meat remained high even though the use of ESC and other antibiotics in chicken farms had been reduced [5]. Also, a high prevalence of cefotaxime-resistant *E. coli* was reported for samples of chicken meat (74.9% positives) and turkey meat (40.1%), which had been obtained in different parts of Germany between 2012–2013 [6].

The increasing demand for poultry products has led to intensive broiler breeding practices at the risk of increasing antibiotic consumption with the ensuing selection for resistant bacteria [7]. Intensive poultry farming may also lead to emissions of ESC-resistant bacteria into the surrounding environment, e.g., through wastewater, air and dust, transport devices, human carriers, etc. [8,9,10]. The ability to remain viable in the environment could further contribute to the environmental transmission of ESC-resistant *E. coli* [11]. The fact that determinants of ESC-resistance, i.e., extended spectrum β-lactamases (ESBL) or plasmid-mediated AmpC (pAmpC), have remained in food animal production systems despite the cephalosporin ban has been suggested to be related to a continuous influx of new resistant *E. coli* clones from the import of poultry from countries still using the antibiotics in their production systems [12]. However, the persistence of clones in the production system could also be due to environmental adaption, co-selection of ESC-resistance through use of related antimicrobials, or even cross-resistance to other distinct antimicrobial classes or biocides [13].

Common ESC-resistance genes found in European food animals include *bla*_CMY-2_, *bla*_CTX-M_, *bla*_TEM-52_, and *bla*_SHV-12_. In Spain, *bla*_CTX-M_-producing *E. coli* could be isolated from 59.1% of broiler chickens [14]. In the Netherlands, a study [15] reported that 76.8% of chicken meat samples contained ESBL-producing *E. coli*, which carried genes such as *bla*_CTX-M-1_ (58.1%), *bla*_TEM-52_ (14.0%) and *bla*_SHV-12_ (14.0%). In Denmark, ESC-resistant *E. coli* containing *bla*_CMY-2_, *bla*_SHV-2_ or *bla*_CTX-M-1_ were isolated in a survey of Danish broiler production systems and associated slaughterhouses between 2009–2011, and the finding of *E. coli* with similar ESC-resistance genes on both farms and slaughterhouses was taken to indicate vertical transmission [12]. Using whole genome sequencing (WGS), Pietsch and colleagues [16] showed that 164 *bla*_CMY-2_ producing *E. coli* isolates, which came from humans, livestock, and (poultry) meat products, belonged to 59 different sequence types (STs) with several STs occurring in both humans, livestock and foods thus highlighting the diversity of genetic backgrounds for carriage of *bla*_CMY-2_ and risk of transmission between animals and humans. Clearly, WGS presents an opportunity to investigate circulation of dominant ESC-resistance genes in different *E. coli* STs associated with poultry production in Denmark.

The objective of this study was to investigate the prevalence of ESC-resistant *E. coli* in the production of broilers from farms to slaughterhouses in Denmark and use WGS to elucidate the potential transmission of ESC-resistant *E. coli* isolates in different parts of the poultry production pyramid.

## 2. Results

### 2.1. Prevalence of ESC-Resistant E. coli in the Environment of Broiler Production Systems

From the conventional Production System 1 (Table 1), in total, 82 environmental samples were collected from farm sites of egg producers, hatcheries, and slaughter chickens in 2015 and 2016, whereas 175 environmental samples from the central breeding were obtained in 2015. Only seven (prevalence of 2.7%, 95% CI 1.2–5.8%) of the 257 samples tested positive for the presence of ESC-resistant *E. coli*. No ESC-resistant *E. coli* positive samples were found in the environment of hatcheries or facilities rearing slaughter chickens. In the central breeding, ESC-resistant *E. coli* positive samples (i.e., two samples) originated from one swab sample of transport boxes and one sock sample obtained four weeks after the introduction of the imported parent poultry chicks in the farm. The remaining five ESC-resistant *E. coli* positive samples came from the egg producing farms, specifically three positive samples came from sock samples that were obtained four weeks after the introduction of mature hens and cocks into the two houses. Of the two positive samples obtained after eight weeks, one came from a house that also tested positive after four weeks while the second positive sample came from a different house.

From Production System 2 (Table 1), 38 samples were tested, where 15 samples came from the conventional production and 23 from the organic production. In the conventional production, four samples were positive (prevalence of 26.7%, 95% CI 8.9–55.17%) for the presence of ESC-resistant *E. coli*, with two positive samples coming from the central breeding facility four weeks after the introduction of imported poultry chicks, and two positive sock samples found in the egg producing facility four weeks after the introduction of breeding parent poultry birds. In the organic production, 13 (56.5%, 95% CI 34.9–76.1%) out of the 23 samples tested positive for ESC-resistant *E. coli*. Of these positive samples, two were found in the central breeding farm 16 weeks after the introduction of imported chicks, four came from the hatchery four weeks after the introduction of eggs, and lastly, seven came from three different farms with slaughter chicken production.

### 2.2. Prevalence of ESC-Resistant E. coli in the Slaughterhouses

The slaughterhouse sampling program revealed an overall prevalence of ESC-resistant *E. coli* of 18.9–29.5% and 15.7–20% for thigh and intestinal samples, respectively. Specifically in Slaughterhouse A, 18 (20%) of the 90 intestinal samples contained an ESC-resistant *E. coli* isolate, while 28 (18.9%) of the 148 thigh samples tested positive (Table 2). In Slaughterhouse B, 33 (15.7%) of the 210 intestinal samples contained ESC-resistant *E. coli*. The prevalence of ESC-resistant *E. coli* was higher in thigh samples with 59 (29.5%) of 200 samples testing positive in the qualitative presence/absence test.

Enumeration of ESC-resistant *E. coli* in intestinal and thigh samples obtained from the slaughterhouses during the first 4 months of 2018 revealed that levels exceeded the detection limit of 100 CFU/g in 126 (36.9%) of the 342 samples (Table 3). Fifteen (7.5%) of the two hundred intestinal samples from the slaughterhouses contained more than 10^3^ CFU/g, while none of the 142 thigh samples exceeded a content of 10^3^ ESC-resistant *E. coli* per g. As for the overall contamination levels in Slaughterhouse A, 45% of samples contained more than 100 ESC-resistant *E. coli* per g. For Slaughterhouse B, 42 (34.4%) of 122 samples contained more than 100 ESC-resistant *E. coli* per g.

### 2.3. Multi-Locus Sequencing Types of the ESC-Resistant E. coli Isolates

A total of 162 ESC-resistant *E. coli* isolates were obtained, with 24 of these coming from ESC-positive samples from the production systems (seven isolates from Production System 1 and 17 isolates from Production System 2, Table 1), while the remaining 138 isolates came from the two slaughterhouses (46 isolates from Slaughterhouse A and 92 isolates from Slaughterhouse B, Table 2).

In the two poultry production systems, analysis of the sequence types (STs) revealed that ST429 dominated in Production System 1 (n = 7, 100%), while ST155 (n = 16, 94.1%) and ST1196 (n = 1, 5.9%) were found in Production System 2 (Table 1).

MLST typing of the 138 ESC-resistant *E. coli* isolates from the slaughterhouses showed the presence of a diverse set of STs in that environment. Looking at Slaughterhouse A, eight known STs were represented with the following distribution: ST2040 (n = 33, 71.7%), ST429 (n = 4, 8.7%), ST23 (n = 2, 4.3%), ST1818 (n = 2, 4.3%), ST1011 (n = 1, 2.2%), ST1196 (n = 1, 2.2%), ST162 (n = 1, 2.2%), ST580 (n = 1, 2.2%), and one strain not being assigned to a known MLST (Table 4). Slaughterhouse B also contained a diverse set of ESC-resistant *E. coli* from 10 different STs (Table 5), namely ST2040 (n = 52, 56.6%), ST429 (n = 11, 12%), ST23 (n = 8, 8.7%), ST1286 (n = 6, 6.5%), ST4663 (n = 6, 6.5%), ST10 (n = 3, 3.2%), ST115 (n = 3, 3.2%), ST57 (n = 1, 1.1%), ST350 (n = 1, 1.1%), and ST117 (n = 1, 1.1%). Taken together, among 162 ESC-resistant *E. coli* isolates, 15 known STs were identified, with ST2040 (n = 85, 52.5%) and ST429 (n = 22, 13.6%) being the common STs, which could be isolated in both slaughterhouses. ST429 was also isolated in Production System 1. One of the 162 strains belonged to an unknown ST.

### 2.4. Resistance and Virulence Genes in ESC-Resistant E. coli Isolates

A closer look at the diversity of ESC-resistance genes in the *E. coli* isolates revealed seven different types of β-lactamases (*bla*_CMY-2_, *bla*_CTX-M-1_, *bla*_CTX-M-14_, *bla*_SHV-12_, *bla*_TEM-52B_, *bla*_TEM-52C_, and the mutation in the promoter of ampC) (Figure 1).

All seven ESC-resistant *E. coli* (ST429) from Production System 1 contained the *bla*_CMY-2_ resistance gene. In Production System 2, the 16 ESC-resistant *E. coli* ST155 isolates contained the *bla*_TEM-52B_ gene, while the ST1196 isolate came from an organic slaughter chicken producer and contained *bla*_CTX-M-14_ (Table 1).

In Slaughterhouse A, the dominant ESC-resistant *E. coli* ST2040 isolates (thighs n = 20, intestinal n = 13, Table 4, Figure 1) contained *bla*_CMY-2_. This resistance gene was also found in other STs from Slaughterhouse A, including ST429 (thighs n = 4), and ST162 (intestinal n = 1). Other *E. coli* ESC-resistance genes included *bla*_CTX-M-1_ (ST1818, thighs n = 2), *bla*_SHV-12_ (ST1011, thighs n = 1), and the mutation conferring upregulation of the ampC promoter (ST23, thighs n = 1, intestinal n = 1). One strain with the upregulation of the ampC promoter did not belong to a known ST (Table 4).

In Slaughterhouse B, *bla*_CMY-2_ was found in the following STs; ST2040 (thighs n = 35, intestinal n = 17), ST10 (thighs n = 2, intestinal n = 1), ST57 (thighs n = 1), ST350 (thighs n = 1), ST429 (thighs n = 8, intestinal n = 3) and ST1286 (thighs n = 4, intestinal n = 2) (Table 5). The other ESC-resistance genes were represented by *bla*_CTX-M-1_ (ST117, intestinal n = 1), *bla*_TEM-52C_ (ST115, thighs n = 2, intestinal n = 1), and upregulation of the promoter of ampC (ST4663 (thighs n = 2, intestinal n = 4) and ST23 (thighs n = 4, fecal n = 4)).

An analysis of 162 whole genome sequences of ESC-resistant *E. coli* isolates showed that they also contained determinants for resistance to other antibiotics such as aminoglycoside (*aac*(3)-*Vla*, *aadA*1), macrolides (*mdfA*), sulphonamide (*sul1*), and tetracycline (*tetA*).

Several virulence factors were detected among the *E. coli* isolates including *lpfA*, which encodes a long polar fimbria, in 126 (77.8%) of the *E. coli* isolates (Appendix A). The prevalence of *iss*, which promotes *E. coli* immune evasion by increasing serum survival, was 139 (85.8.7%). Eighty-eight (54.3%) isolates harbored *cma*, which encodes for colicin. Two kinds of temperature-related virulence genes were detected, *tsh* (temperature-sensitive hemagglutinin) in 92 (56,8%) isolates and *astA* (EAST-1 heat-stable toxin production) in 28 (17.2%) isolates. The glutamate decarboxylase *gad* gene was found in 98 (60.5%) isolates, while the ATP-binding cassette (ABC) transporter *mchF* gene was detected in 33 (20.4%) isolates. Six (3.7%) of the *E. coli* isolates carried the *iha* gene for adherence to host cells, while more rare genes included *vat* (1, 0.6%, vacuolating autotransporter toxin production), *pic* (1, 0.6%, serine protease autotransporters of Enterobacteriaceae), *air* (4, 2.5%) and finally *eilA* (4, 2.5%), which is associated with enteroaggregative *E. coli* pathotype.

### 2.5. cgMLST Analysis

The neighbor-joining tree that was constructed based on a core genome multi-locus sequence typing (cgMLST) scheme with 2513 alleles showed the genetic relatedness among the 162 ESC-resistant *E. coli* isolates from the two poultry production systems and two slaughterhouses (Figure 1). Major branches formed in the cgMLST-based neighbor-joining tree coincided with the STs obtained from the seven-gene MLST scheme. The tree revealed that highly similar ESC-resistant *E. coli* isolates from the same cgMLST, e.g., ST2040 (allele differences 0–34), could be isolated at both slaughterhouses. As for the second most abundant cluster, which was comprised of members of the ST429 (allele differences 0–12), similar isolates occurred in samples from Production System 1 (inner ring; yellow) as well as in samples from the two slaughterhouses (inner ring; A—red; B—blue), indicating their transmission through the production chain. Also, highly similar isolates from ST1196 were isolated from Production System 2 (inner ring; green) and Slaughterhouse A (red). The ST155 (allele differences 0–19) isolates from Production System 2 (inner ring; green) made up a separate cluster that was not found elsewhere.

### 2.6. Plasmid Analysis of ESC-Resistant E. coli Strains with bla_CMY-2_

Since *bla*_CMY-2_ was the most prevalent ESC-resistance gene (found in 73% of the isolates) in this study, the genetic relatedness (cgMLST) and plasmid incompatibility types were further analyzed for the 119 *bla*_CMY-2_ positive isolates (Figure 2). Only two incompatibility groups were found, with IncI1 being the most common (n = 86), while IncK was found in 32 isolates. Notably, the IncK plasmids were found in isolates belonging to several MLSTs, i.e., ST429 (n = 22), ST1286 (n = 6), ST162 (n = 1), ST2040 (n = 1), ST57 (n = 1), and ST350 (n = 1) (Figure 2). In contrast, *bla*_CMY-2_ producing isolates harboring the IncI1 plasmids belonged to ST2040 (n = 83) or ST10 (n = 3). The plasmid type of one isolate belonging to ST2040 could not be determined using the Illumina sequence data.

## 3. Discussion

This study investigated the occurrence of ESC-resistant *E. coli* in two poultry production systems and two slaughterhouses in Denmark during the period of 2015–2018. Of the 943 environmental, fecal and thigh samples that were tested qualitatively, an overall 17.2% (95% CI 14.8–19.7%) tested positive and led to the isolation of 162 ESC-resistant *E. coli*. Third generation cephalosporins were not in use in the country during this period, meaning that the presence of the ESC-resistant bacteria was not related to direct antibiotic use. Other studies have also documented the persistence of ESC-resistant *E. coli* despite limited or no use of antibiotics including cephalosporins [5,17,18].

Two integrated production systems, which operated as a production pyramid from the rearing of parent birds to broilers ready to go to the slaughterhouse, both contained ESC-resistant *E. coli*, but with different prevalences of 2.7% (95% CI 1.2–5.8%) and 44.7% (95% CI 29–61.5%), respectively. Concurrent studies in other European countries reported the isolation of ESC-resistant *E. coli* in 100% of surveyed Dutch and Italian poultry farms [19,20], while in our study, 10 of 33 (30%) tested farms tested positive (Table 1). The current study did not investigate factors, which could explain these differences; however, the use of other antibiotics, dietary supplements, and sanitation practices are known to influence the prevalence and transmission of ESBL/pAmpC-producing *E. coli* in poultry pyramids [1,21].

As shown in Table 1, the positive samples primarily came from the top of the pyramid (central breeding (1.1% [95% CI 0.2–4.5%] in Production System 1 and 44.4% [95% CI 15.3–77.3%] in Production System 2) and egg producers (9.3% [95% CI 3.5–21.1%] in Production System 1 and 50% [95% CI 22.3–77.8%] in Production System 2)), indicating either repeated introduction of these ESC-resistant *E. coli* from an external source or that they had colonized the environment in the top of the system. Several studies also observed a higher prevalence at the top of the poultry pyramids in countries such as Sweden, Italy, and Finland with prevalence ranging from 92.5% in Italian parent stock chicks to 26.7% in Finnish parent birds [17,21,22]. An earlier study conducted during 2010 and 2011 in Denmark reported a prevalence of 93% in 29 broiler parent farms [12], which is markedly higher than results from the present study of 2015–2016 samples, indicating an improvement. Nevertheless, continued initiatives to eradicate antibiotic resistant bacteria at the top of the pyramid, i.e., in the farms that produce the grandparents, would be key to future prevention [17].

At the lower levels of the poultry pyramid (i.e., hatchery and slaughter chickens) all samples tested negative for ESC-resistant *E. coli* in the two conventional production systems. In contrast, the occurrence of ESC-resistant *E. coli* in 50% (95% CI 24.0–76%) of samples from one hatch of the organic slaughter chickens in Production System 2 suggested intermittent, vertical, poor hygiene and/or external environmental contamination could play a role. The decreasing trend through the poultry production pyramid was also observed in the study of Projahn and colleagues [23], who reported that the rate of colonization of hatchlings was low or variable despite the finding of ESC-resistant enterobacteria or *E. coli* in parent flocks and environmental samples. The disappearance of a specific type of ESC-resistant *E. coli* (*bla*_CMY-2_) through the pyramid was also reported in Finland [22], indicating the specific strain was not competitive during rearing of the chicks.

The organic part of Production System 2 carried a higher load than the conventional part (56.5% vs. 26.7%). This result contrasted with a recent report of the prevalence of ESC-resistant *E. coli* in organic systems being lower compared to a conventional system in Italy [24,25]. A similar observation was also reported from Germany [26]. Additional studies are required to elucidate if the ESC-resistance of *E. coli* varies systematically between conventional and organic production systems including possible roles of exposure to the external environment for organic chickens.

*Escherichia coli* with resistance to ESC antibiotics occurred in around 20% of the broiler samples from the two Danish slaughterhouses (Table 2), even though a voluntary ban on the use of 3rd generation cephalosporins on poultry farms had existed for 7–8 years. Looking at data extracted from the DANMAP surveillance program in Denmark, the country-wide prevalence of ESC-resistant (reported as ESBL/AmpC) *E. coli* in broilers obtained at slaughterhouses was 15–16% in 2016–2018 [3] (see Appendix A). A French study also found that, despite reduced antimicrobial use in chicken production, a high prevalence (91.7%) of ESC-resistant *E. coli* was found in 48 poultry samples [5]. Recent DANMAP 2020 surveillance data do, however, indicate that the ban may have contributed to reducing the prevalence of these resistant bacteria to just 3% in samples of Danish broilers randomly taken in slaughterhouses (Appendix A).

The load of ESC-resistant *E. coli* was above 100 CFU/g in >50% of caecum samples, confirming the intestinal carriage of the bacteria. Thigh contamination levels were lower in Slaughterhouse B (95% < 100 CFU/g) compared to Slaughterhouse A (60% < 100 CFU/g), indicating fewer cross-contamination events in the first slaughterhouse. Most of the positive thigh samples contained between 10^2^ and 10^3^ CFU/g. Similar concentrations of ESC-resistant *E. coli* in broilers have been reported for Italian, Dutch, and German slaughterhouses [20,27,28]. These findings strongly indicated that cross-contamination between the broiler meat and the intestinal tract happened during the slaughter process, thus agreeing with observations by Projahn et al. [29]. The difference in thigh contamination levels between the two slaughterhouses may be due to factors not examined in this study, including variation in sanitation programs, hygienic design of process lines, etc.

A total of 162 ESC-resistant *E. coli* were isolated during this study, with 24 coming from the two production systems and 138 coming from the two slaughterhouses. All seven isolates from Production System 1 belonged to ST429 and harbored the *bla*_CMY-2_ gene, while 16 of 17 isolates from Production System 2 belonged to ST155 with *bla*_TEM52-B_ and 1 to ST1196 with *bla*_CTX-M-14_. Fifteen STs with six different resistant genes (*bla*_CMY-2_, *bla*_CTX-M-1_, *bla*_CTX-M-14_, *bla*_SHV-12_, *bla*_TEM-52C_, upregulation of ampC) were found in the two slaughterhouses, indicating a large diversity of β-lactamases in the slaughterhouses. This finding is similar to what has previously been reported and thought to be caused by cross-contamination events in the slaughterhouse from in-house established and repeatedly introduced types of ESC-resistant *E. coli* from incoming flocks [27,28]. The prevention of cross-contamination in the slaughterhouses could be based on the screening of resistance to selected antibiotics in flocks prior to the arrival at the slaughterhouse such that resistant flocks can be processed last just prior to the daily cleaning and disinfection routine. Initiatives aimed at improving hygiene, especially at the evisceration step, would also decrease the transfer of resistant bacteria from the intestinal tract to the meat [27].

ST2040 with *bla*_CMY-2_ dominated in both slaughterhouses with a prevalence of 54% among ESC-resistant *E. coli* in this study. The presence of this ST has recently been observed in countries such as Germany [16], Norway [30] and Finland [22].

*E. coli* ST429 carrying *bla*_CMY-2_ was the second most abundant and found in Production System 1 and both slaughterhouses. Mo et al. [30] also reported finding ST429-*bla*_CMY-2_ in four of 10 Norwegian broiler farms in 2016, while the same ST appeared on Finnish farms [22]. ST429-*bla*_CMY-2_ has also been found in the Italian broiler production [31].

*bla*_CMY-2_ is an important pAmpC gene in ESC-resistant *E. coli* [32,33]. Our findings showed a greater proportion of *bla*_CMY-2_ (73%) compared to the 32–48% observed among ESC-resistant *E. coli* in an earlier study of poultry meat [28,34]. Moreover, carriage of *bla*_CMY-2_ has been observed in a diverse set of STs, i.e., 59 different STs in German broiler and chicken meat [16], including ST10 and ST38 which were found to circulate in Scandinavian poultry from 2010–2014 [12,18,35]. Several *bla*_CMY-2_ STs were also reported in the poultry production pyramid in Switzerland [2], Italy [31] and Spain [36].

ESC-resistant *E. coli* belonging to ST155 and harboring *bla*_TEM-52B_ were found in Production System 2 but not in the slaughterhouses (Figure 1). However, an ST115 *E. coli* containing *bla*_TEM-52C_ was found in Slaughterhouse B. Both STs harboring *bla*_TEM-52-group_ are known to be present in the Danish broiler system [37].

The mutation conferring upregulation of chromosomal ampC expression was found in 12% of the ESC-resistant *E. coli* isolates in our study. Similar *E. coli* with mutational upregulation of chromosomal ampC were also detected in Finnish chicken farms, indicating a common source or frequent occurrence in poultry production [38].

Besides the AmpC/ESBLs resistance genes, the ESC-resistant *E. coli* carried additional genes encoding antibiotic resistance including: aminoglycoside resistance (*aac(3)-Vla*, *aadA1*), macrolide resistance (*mdfA*), sulphonamide (*sul1*), tetracycline resistance (*tetA*). Previous studies conducted in Spain and Finland have similarly demonstrated carriage of additional resistance genes resulting in ESC-resistant *E. coli* with resistance to multiple antibiotics [22,36].

The present study used WGS to provide a more detailed insight into the genetic relatedness of ESC-resistant *E. coli* in the broiler production system in Denmark. Using the ad hoc 2513 alleles cgMLST scheme, the genetic relationship among ST429 with *bla*_CMY-2_ was observed to be close (Figure 1), thus supporting the vertical transmission of this clone. ST2040 isolates carrying *bla*_CMY-2_ from the two slaughterhouses were highly similar, perhaps indicating a common source in 2017–2018. A common contamination source was also hypothesized to be the cause of the presence of genetically similar ST2040 isolates in Norwegian chicken meat samples obtained in 2016 [39].

The predominant presence of ESC-resistant *E. coli* in the top of the production pyramid has raised suspicions of there being a common source of ESC-resistant *E. coli*. Earlier investigations have noted that this common source consists of imported birds coming from the grandparent flocks, pointing to direct vertical transmission from generation to generation [12,17,18]. Reasons for the maintenance of ESC-resistance may be due to use of cephalosporins in grandparent flocks or co-selection by other unknown environmental factors [40]. Moreover, adding to the evidence of a common source is the observation that the ST429 from the top of Production System 1 resembled the ST429 found in Finnish farms [22], which may have imported chicks from the same grandparent flocks.

*bla*_CMY-2_, this study’s most common pAmpC gene, has been identified on IncI1, IncK, IncA/C plasmids and even sporadically on IncX and IncHI2 plasmids in *E. coli* in European livestock production [16,41,42,43,44]. However, only IncI1 (n = 87) and IncK (n = 31) plasmids carrying *bla*_CMY-2_ were present in our study. Earlier studies also reported *bla*_CMY-2_ being associated with IncI1 and IncK plasmids in *E. coli* isolated from Dutch and Danish poultry meat [45,46] from 2006–2012. The risk of zoonotic transmission of *bla*_CMY-2_ in Denmark was highlighted in 2016 by the finding of genetically similar IncI1 *bla*_CMY-2_-producing *E. coli* isolates in a human clinical case and broilers [47]. The finding of ESC-resistant *E. coli* in humans with a putative link to contaminated broilers is worrisome as these resistant bacteria may cause common cephalosporin treatments of human infections to fail [2].

## 4. Materials and Methods

### 4.1. Sampling Strategy

To survey the prevalence of ESC-resistant *E. coli* in all segments of the rearing and slaughter of the broilers in Denmark, samples (sock, swab, and dust samples) were collected from the production environment of two production systems during 2015 and 2016 for a total of 295 samples obtained from 33 farms. Sampling sites included all steps along the integrated production chain for each production system, i.e., the central breeding farm (rearing of imported chicks to mature parent birds, i.e., parents of the slaughter chickens), sites for egg producers (mature parent birds transferred for production of fertilized eggs), hatcheries (hatching of chicks), and rearing farms (growth of the chicks into full-sized slaughter chickens) (Additional Appendix A). In 2015, only samples from central breeding farms were collected, whereas all segments of the production system were sampled in 2016. Production System 1 has conventional production of slaughter chickens (i.e., broilers), while Production System 2 produces both traditional and organic slaughter chickens. During 2017 and the first four months of 2018, monthly samples of broiler thighs and intestines (caeca) were collected from two slaughterhouses (A and B). The slaughterhouses process slaughter chickens from Production Systems 1 and 2. All samples were refrigerated (5 °C) and transported to the Microbiology laboratory of the Danish Veterinary and Food Administration (DVFA) in Ringsted, Denmark, for further analysis as described in Section 4.4 and Section 4.5. Microbiological analysis was initiated within 24 h of sampling.

### 4.2. Sample Collection in the Broiler Production Systems

In central breeding farms, the production environment in each house was sampled by swabbing equipment in the stables using ten swabs for each sample. The ten swabs were divided into two samples by placing five swabs into sterile stomacher bags. Transport crates were similarly sampled using ten swabs that were subsequently pooled into two samples in sterile stomacher bags. Sock samples were obtained at different times (Table 1) in each house, where farm workers walked through the facility equipped with boots coated with sterile gauze socks to collect bacteria. Two to six pairs of socks were pooled in two samples. The egg-producing farms were sampled using swabs and boot socks to sample the houses (conveyor belts, drinking and feeding systems, surfaces in egg rooms, and floors). In hatcheries, swab samples were taken from the paper coating lining the inside of the trays. Swabs from five trays were pooled in sterile stomacher bags. Dust from four trays (with eggs) was collected in sterile stomacher bags and pooled into one sample. Samples were collected one and four months after hatching had started. Farms, where rearing of the chicks into full-sized slaughter chickens take place, were sampled 15–21 days before slaughter using six pairs of socks, which were subsequently pooled into two samples (three pairs in each) and placed in sterile stomacher bags.

In the case of Production System 1, six central breeding farms (3–5 houses on each farm) were sampled as described above, with repeated visits to five out of the six breeding farms. A total of six egg-producing farms (2–3 houses each) within this production system were visited. Two hatcheries belonging to Production System 1 were sampled. Eight slaughter chicken rearing farms (1–2 houses each) from Production System 1 were sampled once each. For Production System 2, the central breeding farms were sampled four times for the conventional and one time for the organic facility (Table 1). The egg-producing farm in Production System 2 is located on the same site as the central breeding farm and was visited twice. One hatchery belonging to Production System 2 was sampled once. Five slaughter chicken rearing farms, which operated as organic, were visited once each.

### 4.3. Sample Collection in the Slaughterhouses

Two Danish slaughterhouses (Slaughterhouse A and B) were investigated by monthly sampling of chicken carcasses randomly collected by slaughterhouse personnel. For qualitative analysis of the prevalence of ESC-resistant *E. coli*, a total of 648 samples were collected in 2017 and during the first four months of 2018, representing 202 broiler flocks and 58 Central Husbandry Register (CHR) numbers (*ca.* 58 slaughter chicken farms). A total of 238 samples were collected from Slaughterhouse A, including 90 fecal samples from the intestines (caeca) and 148 from the thighs. From Slaughterhouse B, 410 samples were collected. Of these, 210 were from intestines and 200 from thighs.

The load of ESC-resistant *E. coli* was quantified in 342 samples obtained from January to April 2018, as described in Section 2.5. Of these, 220 were collected at Slaughterhouse A (120 from the intestines, and 100 chicken thighs), while 122 samples (80 from intestines and 42 chicken thighs) were collected at Slaughterhouse B. Each intestinal and thigh sample consisted of the pooled content of material from five broilers. Samples were collected in sterile stomacher bags and processed for microbiological analysis as described below in Section 4.4 and Section 4.5.

### 4.4. Qualitative Detection of ESC-Resistant E. coli

For environmental samples from the production systems, two hundred and twenty-five mL of 0.1% (*w*/*v*) buffered peptone water (BPW, Oxoid, Basingstoke England) was added to each stomacher bag followed by gentle mixing. Samples were incubated at 37 °C for 18–22 h and subsequently streaked onto MacConkey agar plates (Oxoid, Basingstoke, England) containing 1 mg/L of cefotaxime (CTX, Sigma, Steinheim, Germany) using a 10 μL inoculation loop to ensure semi confluent growth according to the EURL standard procedure [48]. One presumptive ESC-resistant *E. coli* colony, typical red/purple, was picked and identified as *E. coli* by re-streaking on CHROM Orientation agar (Becton Dickinson A/S, Brøndby, Denmark). Each presumptive ESC-resistant *E. coli* isolate was sub-cultured in 1 mL of Luria–Bertani (LB) broth (Oxoid) at 37 °C for 18–22 h, after which glycerol was added to a final concentration of 18% (*v*/*v*) before freezing the tube at −80 °C for long term storage.

For slaughterhouse intestinal samples, the presence of ESC-resistant *E. coli* was analyzed by cutting the five intestines open using sterilized scissors and pooling the content in a new stomacher bag. BPW (0.1%) was added at a ratio of 1:9 (*w*/*w*), and samples were incubated for 18–22 h at 37 °C. Presumptive ESC-resistant *E. coli* colonies were isolated, identified and preserved for further analysis as described above. For thigh samples, 225 mL 0.1% BPW were added to stomacher bags containing five chicken thighs. Samples were incubated and analyzed as described above, followed by isolation and preservation of presumptive ESC-resistant *E. coli*.

### 4.5. Quantitative Enumeration of ESC-Resistant E. coli in Slaughterhouse Samples

Pooled intestinal samples (approx. 5 g) comprised of the content of the caeca from five broilers were prepared as detailed as above, and the content was placed in sterile stomacher bags. The pooled samples were weighed, diluted 1:9 with 0.1% BPW, and mixed in a stomacher for 1–2 min. For thigh samples, five thighs were pooled, weighed, and diluted 1:9 with 0.1% BPW in a stomacher bag and shaken vigorously for 1–2 min. Ten-fold dilutions were made in cold 0.9% (*w*/*v*) saline until the desired final dilution (10^−4^). Aliquots of 100 μL from suitable dilutions were spread plated onto MacConkey plates containing 1 μg/mL cefotaxime. Agar plates were incubated for 18–22 h at 44 °C followed by the enumeration of the number of typical red/purple colonies as described in the European Union Reference Laboratory for Antimicrobial Resistance protocol [49]. Additional verification of individual colonies from countable plates was performed by streaking colonies on CHROM Orientation agar. Confirmed colonies (1–3 from each plate) were re-streaked on MacConkey with 1 μg/mL cefotaxime, grown overnight at 37 °C, followed by transfer of colony mass into 1 mL of LB broth with 18% glycerol for long-term storage at –80 °C.

### 4.6. Whole Genome Sequencing Analysis

In preparation for whole-genome sequencing (WGS) on an Illumina MiSeq platform (Illumina, San Diego, CA, USA), all 162 ESC-resistant *E. coli* isolates were grown on Tryptic Soy blood agar (Statens Serum Institut, Copenhagen, Denmark) and incubated for 24 h at 37 °C at the Center for Genomic Epidemiology (CGE), National Food Institute, Technical University of Denmark. Single colonies were sub-cultured in Luria–Bertani broth (Oxoid) for 18 h at 37 °C. Bacterial DNA was extracted from this overnight culture using the Easy-DNA kit (Invitrogen, Waltham, MA, USA) yielding concentrations of 0.18–0.28 ng/μL. Sequencing sample libraries were constructed using the Illumina Nextera XT prep kit (Illumina, San Diego, CA, USA) following the standard WGS library protocol (as per the manufacturer’s instructions), and sequenced with the V2 NextSeq mid output flow cell (2 × 150 bp paired-end reads). Raw reads were subjected to quality control (FastQ quality control tool, https://www.bioinformatics.babraham.ac.uk/projects/fastqc/, accessed on 1 March 2020) with removal of reads with a quality score below 20 and de novo assembly of remaining reads into draft genomes using the pipeline of CGE [50]. The assembled FASTA sequences were then analyzed using CGE bioinformatics tools to determine the sequence type, antimicrobial resistance genes/fluoroquinolone resistance-associated mutations, virulence genes, plasmid types, and plasmid MLST (pMLST) by using the following pipelines: MLST Finder 2.0, Resfinder 2.1, Virulencefinder 2.0, Plasmidfinder 2.1 and pMLST 2.0 [51,52,53] (https://cge.cbs.dtu.dk/services/cge/, accessed on 15 August 2022). In silico MLST of *E. coli* isolates was determined based on the allelic variation in seven housekeeping genes (*adk*, *fumC*, *gyrB*, *icd*, *mdh*, *purA*, and *recA*) [54] using a setting of 100% match to assign isolates to known MLST groups, while those isolates without perfect matches were assigned as belonging to unknown MLSTs. For analysis of each isolate with Resfinder 2.1 [55], Virulencefinder 2.0, and Plasmidfinder 2.1, a threshold of 95% for minimum identity was selected together with a minimum length of the contig of 60%. 

### 4.7. Phylogenetic Analyses

The phylogenetic diversity of ESC-resistant *E. coli* genomes was examined through an ad hoc core genome multi-locus sequence-typing scheme (cgMLST) and performed with the Seq-Sphere platform (v5.0.1, Ridom GmbH, Munster, Germany) and a gene-by-gene allele comparison method [56]. The cgMLST pipeline used *E. coli* K-12 as the reference genome, which led to the extraction of 2513 genes in the core genome. This core genome was compared to the assembled genomes of the 162 ESC-resistant *E. coli* isolates. A cluster alert of 15 cgMLST allele distance and minimum of >90% of good targets were used to detect closely related isolates. Based on the distance matrix describing pairwise allelic differences, a neighbor-joining tree was constructed based on the cgMLST target genes (calculating criteria were pairwise ignoring missing values; % column differences) for all 162 ESC-resistant *E. coli* isolates as well as for the subset of 119 *bla*_CMY-2_ carrying ESC-resistant *E. coli* isolates. The resulting phylogenetic cgMLST comparison-based neighbor-joining tree was annotated and visualized using iTOL (v3.5.4) (http://itol.embl.de/, accessed on 20 March 2023) [57].

### 4.8. Statistical Analysis

The 95% confidence interval (95% CI) of prevalence of ESC-resistant *E. coli* was calculated using exact confidence intervals for a binomial proportion.

## 5. Conclusions

The overall prevalence of ESC-resistant *E. coli* in the broiler production systems and slaughterhouses were 17.2% (95% CI 14.8–19.7%) during the years of 2015–2018 in Denmark, despite no cephalosporin usage. Whole-genome sequencing of 162 ESC-resistant *E. coli* isolates from broilers in Denmark revealed the presence of 15 different STs (and one unknown), with a high diversity in the two slaughterhouses and dominance of ST429 (*bla*_CMY-2_) and ST155 (*bla*_TEM-52-B_) in Production Systems 1 and 2, respectively. The most frequent sequence types, ST2040 and ST429, carried *bla*_CMY-2_ on IncI1 and IncK plasmids, respectively. Our data suggests that clonal transmission of ESC-resistance occurs in the poultry production pyramid. Moreover, the intestinal carriage of ESC-resistant *E. coli* likely fosters cross-contamination events during broiler production and slaughterhouse processing, making poultry an important reservoir for ESC-resistant *E. coli* putatively leading to an unwanted exposure of humans upon handling and consumption of contaminated broiler meat. Future work should investigate interventions to prevent the spread of ESC-resistant *E. coli* from farm to fork.

## Figures and Tables

**Figure 1 antibiotics-12-01602-f001:**
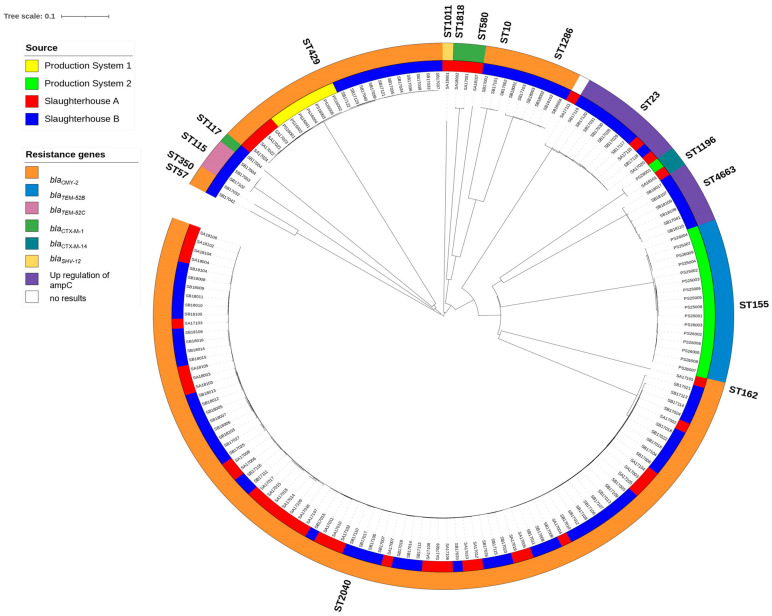
NGS-based neighbor-joining tree of 162 ESC-resistant *E. coli* isolates based on an ad hoc cgMLST including 2513 alleles. The phylogenetic tree was built with SeqShere+ and visualized by iTOL v.6. Different MLST clusters (STs) followed major branches of the cgMLST neighbor-joining tree and are denoted outside the outer ring. The color-coded rings represent the origins of the samples (inner ring) and the different resistance genes in the *E. coli* isolates (outer rings).

**Figure 2 antibiotics-12-01602-f002:**
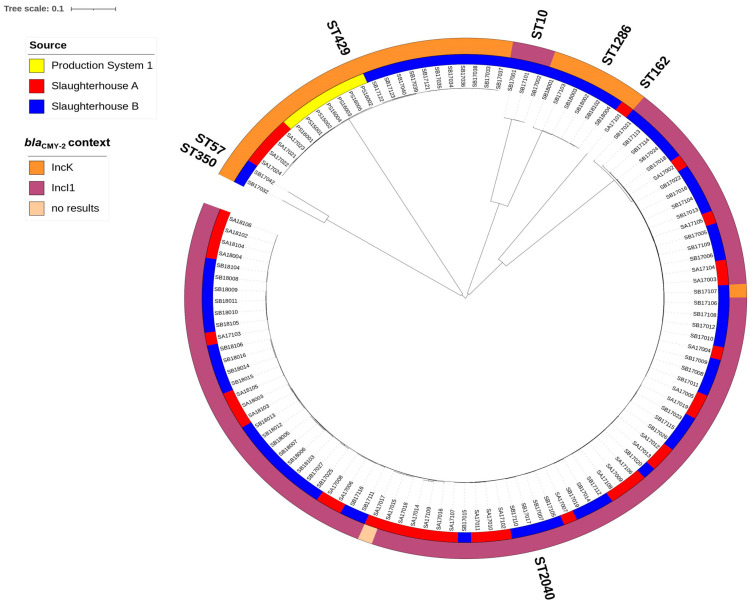
NGS-based neighbor-joining tree of 119 *bla*_CMY-2_ producing *E. coli* isolates based on an ad hoc cgMLST including 2513 alleles. The phylogenic tree was built with SeqSphere+ and visualized by iTOL v.6. Different MLST types (STs) followed major tree branches and are indicated outside the outer ring. The color-coded inner ring represents the source of isolates while the outer ring indicates the plasmid replicon types.

**Table 1 antibiotics-12-01602-t001:** The prevalence of ESC-resistant *E. coli* in the production systems (i.e., 33 farms of which 10 farms tested positive).

		Production System 1		Production System 2 ^c^			
		Conventional (n, Samples)	ESC Pos (%)	Conventional (n, Samples)	ESC Pos (%)	Organic (n, Samples)	ESC Pos (%)
Central Breeding		6 farms (24 houses)		1 farm		1 farm	
	Swab from house	48	0 (0.0)	2	0 (0.0)	NA ^d^	-
	Swab from transport boxes	37	1 ^a^ (2.7)	2	0 (0.0)	NA	-
	Sock samples after 4 weeks	42	1 ^a^ (2.7)	2	2 (100.0)	NA	-
	Sock samples after 16 weeks	48	0 (0.0)	1	0 (0.0)	2	2 (100.0)
	Sub-total	175	2 (1.1)	7	2 (28.5)	2	2 (100.0)
Egg Producers		6 farms (13 houses)		1 farm		1 farm	
	Swab from houses	24	0 (0.0)	2	0 (0.0)	NA	-
	Sock samples after 4 weeks	24	3 ^b^ (12.5)	2	2 (100.0)	4	4 (100.0)
	Sock samples after 8 weeks	6	2 ^b^ (33.3)	2	0 (0.0)	2	0 (0.0)
	Sub-total	54	5 (9.3)	6	2 (33.3)	6	4 (66.7)
Hatchery		2 farms (2 houses)		1 farm		1 farm	
	Hatching	5	0 (0.0)	2	0 (0.0)	1	0 (0.0)
	2. Hatching 4 months later	1	0 (0.0)	NA	-	NA	-
	Sub-total	6	0 (0.0)	2	0 (0.0)	1	0 (0.0)
Slaughter chickens		8 farms (11 houses)		none		5 farms	
	Hatching	14	0 (0.0)	NA	-	14	7 ^e^ (50.0)
	2. Hatching	8	0 (0.0)	NA	-	NA	-
	Sub-total	22	0 (0.0)	NA	-	14	7 (50.0)
Total	All samples	257	7 (2.7)	15	4 (26.7)	23	13 (56.5)

^a^—positive samples ESC-resistant *E. coli* (ST429, *bla*_CMY-2_) from the same farm, ^b^—positive samples (*E. coli* ST 429, *bla*_CMY-2_) from two farms—one being in the same location as the farm in a, ^c^—one house for each of the farm units, all positive ESC-resistant *E. coli* belonged to ST155 (*bla*_TEM-52B_), except one ST1196 (*bla*_CTX-M14_) from one slaughter chicken farm, ^d^—NA—not available (i.e., no sampling), ^e^—positive samples came from 3 different farms.

**Table 2 antibiotics-12-01602-t002:** Qualitative testing of the prevalence of ESC-resistant *E. coli* in intestinal and thigh samples obtained from Slaughterhouse A and B between 2017 and 2018.

Slaughterhouse	Sample Type	Total Number of Samples (n)	Positive Samples (n)	Prevalence (%, 95% CI)
A	Intestine ^a^	90	18	(20.0, 12.6–30.0)
	Thighs ^b^	148	28	(18.9, 13.1–26.4)
B	Intestine ^a^	210	33	(15.7, 11.2–21.5)
	Thighs ^b^	200	59	(29.5, 23.9–36.4)

^a^—each pooled sample consisted of the ceacal content from five poultry carcasses; ^b^—each pooled sample consisted of thighs from five poultry carcasses.

**Table 3 antibiotics-12-01602-t003:** Concentrations of ESC-resistant *E. coli* (CFU/g intervals) in intestinal (caecum) and thigh samples obtained from Slaughterhouse A and B in 2018.

Slaughterhouse	Sample Type	Total Number of Samples	Number of Samples with a Concentration of ESC-Resistant *E. coli* within the Specified CFU/g Interval (10^2^–10^6^ CFU/g)					
			<10^2^	≥10^2^	≥10^3^	≥10^4^	≥10^5^	≥10^6^
A	Intestine	120	61 (50.8) ^a^	51 ^b^	- ^c^	4	3	1
	Thighs	100	60 (60.0)	40	-	-	-	-
B	Intestine	80	40 (50.0)	33	4	2	1	-
	Thighs	42	40 (95.2)	2	-	-	-	-

^a^—the concentration was below the detection limit of 100 CFU/g in the stated number (percent) of samples. ^b^—number of samples which contained between 100 and 999 CFU/g and so on for the other intervals. ^c^—no samples were detected in this concentration interval.

**Table 4 antibiotics-12-01602-t004:** Slaughterhouse A: Multi-locus sequence type (MLST) and occurrence of ESC-resistance genes in 46 ESC-resistant *E. coli* isolates from 148 thighs and 90 intestinal samples.

Genes	MLST-Thighs (n = 28)					MLST-Intestinal (n = 18)						Each ESC-Gene	
	23	429	1011	1818	2040	23	162	580	1196	2040	Unknown	Total	%
*bla* _CMY-2_	- ^a^	4	-	-	20	-	1	-	-	13	-	38	82.6
*bla* _CTX-M-1_	-	-	-	2	-	-	-	1	-	-	-	3	6.5
*bla* _CTX-M-14_	-	-	-	-	-	-	-	-	1	-	-	1	2.1
*bla* _SHV-12_	-	-	1	-	-	-	-	-	-	-	-	1	4.3
*bla* _TEM-52C_	-	-	-	-	-	-	-	-	-	-	-	0	0.0
Up regulation of ampC	1	-	-	-	-	1	-	-	-	-	1	3	6.5

^a^—not found.

**Table 5 antibiotics-12-01602-t005:** Slaughterhouse B: Multi-locus sequence type (MLST) and occurrence of ESC-resistance genes in 92 ESC-resistant *E. coli* isolates from 200 thighs and 210 intestinal samples.

Genes	MLST-Thighs (n = 59)									MLST-Intestinal (n = 33)								Each ESC-Gene	
	10	23	57	115	350	429	1286	2040	4663	10	23	115	117	429	1286	2040	4663	Total	%
*bla* _CMY-2_	2	-	1	-	1	8	4	35	-	1	-	-	-	3	2	17	-	74	80.4
*bla* _CTX-M-1_	- ^a^	-	-	-	-	-	-	-	-	-	-	-	1	-	-	-	-	1	1.1
*bla* _CTX-M-14_	-	-	-	-	-	-	-	-	-	-	-	-	-	-	-	-	-	0	0.0
*bla* _SHV-12_	-	-	-	-	-	-	-	-	-	-	-	-	-	-	-	-	-	0	0.0
*bla* _TEM-52C_	-	-	-	2	-	-	-	-	-	-	-	1	-	-	-	-	-	3	3.3
Up regulation of ampC	-	4	-	-	-	-	-	-	2	-	4	-	-	-	-	-	4	14	15.2

^a^—not found.

## Data Availability

The sequencing data have been deposited with links to BioProject accession number PRJNA1036573 in the NCBI BioProject database (https://www.ncbi.nlm.nih.gov/bioproject/).

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
