# Peer review of "Prevalence and Transmission of Extended-Spectrum Cephalosporin (ESC) Resistance Genes in Escherichia coli Isolated from Poultry Production Systems and Slaughterhouses in Denmark"

_antibiotics, 2023, doi:10.3390/antibiotics12111602_

Round 1

Reviewer 1 Report

Comments and Suggestions for Authors

Dear Authors ,

I have some question 

1- in table 2 the concentration of ESC E.coli (cfu/g) from intestine  it is nomally high as the E.coli is normal inhibitant , how did you differ between pathogenic E.coli and normal one?

2- when you detect the virulence factors of E.coli ,  I observed there is no E.coli carry enterotoxins and shiga toxins , and this for me doesn't make sense .

3-please can you explain what is the methods of ESC resistant transfer between E.coli strains that you isolated in this study?

Author Response

We would like to thank the reviewers for their thoughtful comments. We have strived to address all points raised to hopefully improve the quality of the manuscript.

Below, we have detailed our response to the reviewers’ comments.

Reviewer 1.

Dear Authors ,

I have some question 

1- in table 2 the concentration of ESC E.coli (cfu/g) from intestine  it is nomally high as the E.coli is normal inhibitant , how did you differ between pathogenic E.coli and normal one?

The ESC-resistant E. coli were enumerated on MacConkey agar with 1 mg/L cefotaxime. We were as such not detecting pathogenic E. coli but only focusing on enumeration of E. coli that are resistance this antibiotic at this concentration.

2- when you detect the virulence factors of E.coli ,  I observed there is no E.coli carry enterotoxins and shiga toxins , and this for me doesn't make sense .

We used VirulenceFinder version 2.0 to detect the presence of virulence factors in ESC-resistant E. coli. This data base uses a broad definition of virulence factors including genes involved adhesion, invasion, iron acquisition, etc. We have added more virulence genes to the Supplemental Table S1 and also indicated which category the virulence factor belongs to according to VirulenceFinder. The reviewer is correct that none of the ESC-resistant E. coli contained the genes for shiga toxin or enterotoxins (except for astA in some of the blaCMY-2 and blaCTX-M1 carrying strains). 

3-please can you explain what is the methods of ESC resistant transfer between E.coli strains that you isolated in this study?

We have not investigated the transfer of the ESC-resistance among the E. coli strains from this study. It would be an interesting study to do.

Reviewer 2 Report

Comments and Suggestions for Authors

Dear authors,

Please revise the manuscript as recommended in the attached.

Reviewer.

Comments on the Quality of English Language

Author Response

We would like to thank the reviewers for their thoughtful comments. We have strived to address all points raised to hopefully improve the quality of the manuscript.

Below, we have detailed our response to the reviewers’ comments.

Reviewer 2.

Abstract.

Line 16. Antibiotic was inserted as suggested.

Introduction.

Line 48. In was exchanged for between as suggested.

Line 58. E. coli was inserted. Birds were changed to poultry also as suggested.

Line 69. In was exchanged for between as suggested.

Line 79-89. The final paragraph was condensed as recommended by reviewer 3. 

Results.

Line 93. We respectfully believe that in is the correct preposition here to denote that the sampling took place in 2015 and in 2016.

Line 96. The recommended change was made from (6 samples) to (n=6).

Line 98. Sock samples were used to sample the environment in the stables. As explained in lines 458-460 (Materials and methods), sterile gauze socks were placed on boots of the farm workers, who then walked through the facility to sample the enviroment. The socks were subsequently analyzed for their content of ESC-resistant E. coli using selective enrichment (see section 4.4).

Lines 109 and 112. We have specified that we are talking about poultry.

Line 117. The sequence type information has been added to the revised Table 1.

Line 118. The mistaken placement of the header for section 2.2 has been fixed.

Lines 120-127. We have as suggested made a new table (Table 2) to show the results of the qualitative survey of the prevalence of ESC-resistant E. coli in the slaughterhouses.

Lines 149-151. The information has now been included in Table 1.

Line 159. We have now specified that we found strains belonging to eight known STs.

Line 166. We have noted that we detected one strain that did not belong to a known ST.

Line 189. Figure 1 has been updated to show the names of the betalactamases.  

Lines 190-193. The information has been included into the revised Table 1.

Line 212. We have inserted 162.

Line 230. All prevalences are now listed in Table S1.

Discussion.

Line 276. We have deleted that as requested.

Line 279. We have inserted the missing to.

Line 282. We have made the suggested edits.

Lines 279-287 in the old manuscript. The section has been rewritten and reference to DANMAP surveillance data removed to make the section entirely related to our observations.

Lines 288-293 in the old manuscript. This section has been deleted as suggested.

Line 325. We have made reference to the new Table 2 to make it clearer that we are discussing observations from this study. Respectfully, we believe that it is important to keep this section.

Line 355 (old manuscript). We have made a new subsection to deal with just ST429 CMY-2 to make it clearer how the Norwegian and Finnish findings are relevant for results from this study.

Line 370-371 (old manuscript). The reviewer is correct. We have not investigated mechanisms of the spread of these plasmids. We have deleted the sentence.

Line 372-380 (old manuscript). As suggested by the reviewer, we have shortened this paragraph to focus on the major results.

Line 394 (old manuscript). We have deleted the sentence referring the German study as suggested.

Line 398 (old manuscript). We have reworded the sentence to clarify the similarities between our observations and the observations in Norway, namely that the occurrence of genetically similar E. coli in samples from many difference farms point to a common source of contamination.

Reviewer 3 Report

Comments and Suggestions for Authors

The aimed to investigate the prevalence of resistant E. coli in the chain of production of broilers in Denmark and to perform molecular-epidemiology studies to confirm their hypotheses.

Major issues

In general, the M & M section is not written well. Details are not provided and the text is evasive, I do not know purposedly or inadvertently. I have listed some points herebelow, but there are others throughout the section.

4.1. Please make clear all the sites from where samples were obtained, preferably in a table. Also, please make clear the number of farms visited – the figure of 295 samples is vague.

4.2. Sampling equipment is vague, please specify the type of equipment in each case.

4.3. How was the sampling of the carcasses performed?

The authors must refer to the basic rules of writing a manuscript: M & M need to be so clear to allow future researchers to replicate the work. This is not fulfilled in this manuscript. I will give to the authors the benefit of doubt, but I should say that this recurring model raises suspicions.

Minor issues

The final paragraph of the Introduction should be written in clear wording to make it more easy for understanding by future readers.

The authors should be please the above comments, clarify the issues and improve the manuscript accordingly and thereafter, the evaluation can be continued.

At the moment, I cannot have a recommendation about the manuscript, but only a wish for improvement regarding the points raised.

Author Response

We would like to thank the reviewers for their thoughtful comments. We have strived to address all points raised to hopefully improve the quality of the manuscript.

Below, we have detailed our response to the reviewers’ comments.

Reviewer 3.

Major issues

In general, the M & M section is not written well. Details are not provided and the text is evasive, I do not know purposedly or inadvertently. I have listed some points here below, but there are others throughout the section.

Thank you for this comment. We appreciate your perspective and have strived to improve the M&M section in the revised manuscript. We can assure you that we have not attempted to withhold any important information of relevance for the interpretation of the results.

4.1. Please make clear all the sites from where samples were obtained, preferably in a table. Also, please make clear the number of farms visited – the figure of 295 samples is vague.

We have updated Table 1 to clearly show the number of farms and houses that were sampled in this study. We have also updated the M&M section to correspond to the information in Table 1. The figure of 295 samples represents the total number of samples from the two production systems (all farms and all houses) and can be derived from Table 1 (257+15+23=295). We apologize for the confusion and hope that the additional information is helpful. 

4.2. Sampling equipment is vague, please specify the type of equipment in each case.

We have inserted the missing information in section 4.2 of the revised manuscript.

4.3. How was the sampling of the carcasses performed?

We have specified that the chicken carcasses were randomly picked by the slaughterhouse personnel. The CHR was also noted together with flock information, in total representing 58 CHRs (farms) and 202 flocks.

The authors must refer to the basic rules of writing a manuscript: M & M need to be so clear to allow future researchers to replicate the work. This is not fulfilled in this manuscript. I will give to the authors the benefit of doubt, but I should say that this recurring model raises suspicions.

We sincerely hope that the additional information will help to improve the clarity of our manuscript.

Minor issues

The final paragraph of the Introduction should be written in clear wording to make it more easy for understanding by future readers.

We appreciate the comment and have shortened the objective paragraph to make it easier to read.

 The authors should be please the above comments, clarify the issues and improve the manuscript accordingly and thereafter, the evaluation can be continued.

At the moment, I cannot have a recommendation about the manuscript, but only a wish for improvement regarding the points raised.

Round 2

Reviewer 2 Report

Comments and Suggestions for Authors

Authors,

Attached, I have included my revision recommendations.

Reviewer.

Comments on the Quality of English Language

Can be improved.

Author Response

Response to the reviewers Round 2

We would like to profusely thank the reviewers for taking the time to carefully review our manuscript. Their work has enabled us to make significant improvements to the structure and clarity of the manuscript.

Reviewer 2.

Comments from the yellow labels (all line numbers refer to the round 1 revised manuscript).

L91. Consider improving your result clarity by including a paragraph describing the sample sources and years they were collected.

We have inserted the information that samples were collected from the environment of the broiler production system. The years are stated below in the text.

L. 92. Consider mentioning all production system 1 samplings were done with conventional production samples. “The conventional” has been inserted in front of Production System 1.

L. 92. Consider rephrasing: You mentioned in 2.2 thigh and intestinal samples. Why are they different from the slaughter chicken samples? We have specified that the production systems were analyzed by obtaining samples from the environment from farm sites. As also detailed in the materials and method section, sampling focused on the production environment in farms that produce slaughter chickens.

L. 98. We have, as suggested, added (i.e., two samples).

L 158. 7 was changed to seven, as suggested.

L.174. Consider revising: and one strain belonged to an unknown ST.

The information has been moved to a new sentence: One of the 162 strains belonged to an unknown ST.

L. 533. Inoculation loop was inserted as suggested.

Reviewer 3 Report

Comments and Suggestions for Authors

The manuscript has been vastly improved.
However, the Discussion still needs some further work, as it reads a bit 'shallow'. Please include further ideas in there, e.g., clinical consequences of the findings, therapeutic pathways that need to be followed subsequently to the present findings, practical means for limiting the transfer of AMR within the setting described in the manuscript etc.
After that, the re-revised manuscript can be assessed by the editor only.

Author Response

Response to the reviewers Round 2

We would like to profusely thank the reviewers for taking the time to carefully review our manuscript. Their work has enabled us to make significant improvements to the structure and clarity of the manuscript.

Reviewer 3.

The manuscript has been vastly improved.

We would like to thank the reviewer for this comment.

However, the Discussion still needs some further work, as it reads a bit 'shallow'. Please include further ideas in there, e.g., clinical consequences of the findings,

The reviewer is correct that finding ESC-resistant E. coli in broilers is worrisome. The resistant bacteria can become foodborne and infect human consumers, who then cannot be effectively treated with cephalosporins. We have inserted a sentence to highlight this effect:

L. 400. The finding of ESC-resistant E. coli in humans with a putative link to contaminated broilers is worrisome as these resistant bacteria may cause common cephalosporin treatments of human infections to fail [2].

Therapeutic pathways that need to be followed subsequently to the present findings,

We appreciate the comment but feel it is beyond the scope of this manuscript to speculate on therapeutic pathways. We did not survey the use of antibiotics on the farms.

Practical means for limiting the transfer of AMR within the setting described in the manuscript etc.

An important finding of the study is the “loss” of resistance through the production system. This points to an introduction from the top of the pyramid. AMR free grandparents should therefore be a focus.

In L. 297 of the revised manuscript, we have inserted a sentence to discuss this, reading  “Nevertheless, continued initiatives to eradicate antibiotic resistant bacteria at the top of the pyramid, i.e., in the farms that produce the grandparents, would be key to future prevention [17]”.

In L. 304 of the revised manuscript, we have also highlighted the role of hygiene in the farm environment.

Finally, in L. 349-354 of the revised manuscript, we discuss how antibiotic resistance screening could aid in managing the slaughter of resistant flocks. Also, initiatives aimed at reducing cross-contamination events would likely have an effect.

After that, the re-revised manuscript can be assessed by the editor only.